# Assessment of Pain in Osteoarthritis of the Knee

**DOI:** 10.3390/jpm13071139

**Published:** 2023-07-14

**Authors:** Aricia Jieqi Thirumaran, Leticia Alle Deveza, Inoshi Atukorala, David J. Hunter

**Affiliations:** 1Nepean Hospital, Kingswood, NSW 2747, Australia; 2Sydney Musculoskeletal Health, Kolling Institute, Faculty of Medicine and Health, The University of Sydney, St Leonards, NSW 2065, Australia; 3Rheumatology Department, Royal North Shore Hospital, St Leonards, NSW 2065, Australia; 4Senior Lecturer in Clinical Medicine & Consultant Rheumatologist, University Medical Unit, National Hospital Sri Lanka, Colombo 00700, Sri Lanka; 5Faculty of Medicine, University of Colombo, Colombo 00800, Sri Lanka

**Keywords:** pain assessment, osteoarthritis, self-reported pain scale, clinical trials, quantitative sensory testing

## Abstract

Knee osteoarthritis (KOA) pain is a subjective and personal experience, making it challenging to characterise patients’ experiences and assess their pain. In addition, there is no global standard for the assessment of pain in KOA. Therefore, this article examines the possible methods of assessing and characterising pain in patients with KOA using clinical symptoms, pain assessment tools, and imaging. We examine the current methods of assessment of pain in KOA and their application in clinical practice and clinical trials. Furthermore, we explore the possibility of creating individualised pain management plans to focus on different pain characteristics. With better evaluation and standardisation of pain assessment in these patients, it is hoped that patients would benefit from improved quality of life. At the same time, improvement in pain assessment would enable better data collection regarding symptom response in clinical trials for the treatment of osteoarthritis.

## 1. Introduction

Osteoarthritis of the knee (KOA) is a chronic musculoskeletal condition with a multifactorial aetiology. KOA results in chronic knee pain and disability, impacting an individual’s function and quality of life. As pain is a unique, deeply personal, and subjective experience, it can be difficult to assess the severity and intensity of the pain that an individual experiences [1]. Current KOA management comprises education, weight loss, exercise, analgesia, as well as surgical interventions, such as knee replacements [2]. In this review, we explore the ways to evaluate and quantify pain associated with KOA in a clinically meaningful way in order to better understand the experiences and manage their symptoms more effectively.

### 1.1. Epidemiology, Burden, and Impact of Osteoarthritis

More than 240 million individuals worldwide have symptomatic osteoarthritis, which impacts their functioning [3,4]. In 2013, osteoarthritis was the 13th leading cause of global years lived with disability (YLDs), with a mean YLDs of more than 12.8 million years [4]. In 2019, osteoarthritis associated with high body mass index attributed to 21% of the number of YLDs lost [5]. Australasia had the highest age-standardised YLDs rate in 2019, which has been attributed to the high prevalence of obesity in Australia [5]. The increase in the frequency of osteoarthritis has been primarily attributed to increasing population age and obesity prevalence [6]. The prevalence of KOA is highest around 50 years of age [7].

KOA results in pain and negatively impacts the quality of life, physical function and daily activities of patients [7,8]. In addition, osteoarthritis impacts patient mobility, leading to significant disability [9]. Consequently, KOA is associated with higher levels of psychological distress [7]. Osteoarthritis is associated with depression, anxiety, as well as the risk of substance abuse due to the reliance on analgesics for ongoing pain relief [10]. A retrospective cross-sectional study based on a South Korean national survey involving over 22,000 participants by Lee et al. (2020) [10] found a strong association between poor mental health, reduced quality of life and pain from osteoarthritis, with knee osteoarthritis with ongoing knee pain being a significant contributor to depression and stress in females with knee osteoarthritis [10]. Additionally, Wise et al. (2009) [11] found that poor mental health is associated with worse KOA pain, with worsening mental health increasing the risk of pain flares. The authors found that improvements in mental health correlated with improvements in pain [11]. This further highlights how interlinked mental health and psychological distress are with the pain that patients experience.

KOA poses a significant burden on the healthcare system as it is the main contributor to the rise in the need for joint replacements [12]. The cost of osteoarthritis accounts for up to 2.5% of the gross national product of most developed countries [13]. The cost of medications for osteoarthritis only accounts for approximately 10% of the direct costs. The remainder of the cost is a result of hospital admissions, elective orthopaedic procedures, doctor appointments, and rehabilitation [13]. Additionally, the personal burden on people with KOA cannot be underestimated [12]. The work-related burden of osteoarthritis, as described by Ackerman et al. [7], includes loss of productivity, loss of gross domestic product, and loss of taxation revenue [7]. Osteoarthritis is associated with greater use of sick leave, work limitations, and work loss [7]. In 2012, it was found that more than one in seven adults above the age of 45 in South Sweden had clinically diagnosed KOA with a 2.3% prevalence of knee prostheses [14]. Furthermore, the healthcare burden due to osteoarthritis is projected to increase further. It is estimated that by 2032, there will be more than 26,000 new osteoarthritis consults in the healthcare sector per 1,000,000 population above 45 years of age in Sweden alone [14].

### 1.2. Holistic Assessment of the Person with Osteoarthritis

A thorough history and physical examination are essential for a clinical diagnosis of KOA [15]. In addition, it is vital to assess the pain, function and disability due to KOA. As pain is the main symptom of osteoarthritis, a history of a patient’s pain is integral to establishing a pattern of symptoms consistent with osteoarthritis. Osteoarthritis pain is often described as pain which worsens with exercise and movement. This complicates management as the main non-pharmacological method for the management of osteoarthritis is exercise [6]. Osteoarthritis is also associated with brief morning stiffness (<30 min), especially after sitting or lying down, swelling [7,16,17], as well as reduced range of motion, strength, balance, and proprioception [7]. Some individuals also experience joint locking and giving way [17].

Risk factors of KOA include age, gender, body mass index, abnormal bone shape, such as varus knee, and family history [17]. Additionally, high-impact sports, such as long-distance running, soccer, wrestling, weightlifting [7], and a history of knee injury increase the odds of patients developing KOA [7,18,19].

The impact of KOA pain is pervasive, affecting patients’ lives from all aspects. All aspects and personal or societal implications of osteoarthritis should be addressed when developing a comprehensive and holistic management plan for patients. A holistic biopsychosocial framework (see Figure 1) enables patients’ values and wishes to be considered when assessing someone and formulating a patient-centric plan.

It is imperative to assess the patients’ functional abilities and limitations. This includes evaluating the patient’s ability to walk, squat or kneel, climb stairs, as well as complete activities of daily living. The impact of symptoms on the patient’s ability to participate in work or recreational activities should be scrutinised and discussed [7]. The impact of osteoarthritis on the patient’s life, sexual function, relationships, occupation, hobbies, and sleep should be explored, as well as tailor the management to the needs of the individual patient [17].

Physical examination includes the usual “look, feel, and move” components of joint examination, assessment of neurological function, including strength and sensation, as well as balance and gait assessments. In addition, special tests, such as bulge sign, Lachman’s test, posterior drawer test, collateral ligament integrity, and McMurray test, can be performed if an underlying ligamentous or meniscal injury [17] is suspected.

While the diagnosis can typically be made using clinical features alone, imaging can confirm the diagnosis of osteoarthritis for atypical presentations. Imaging modalities used in KOA include radiographs, Magnetic Resonance Imaging (MRI), ultrasound, and Computerised Tomography (CT) scans. Knee radiographs of frontal (AP or PA) and lateral views, as well as tangential views of the patellofemoral joint, can be performed. Both front and lateral views of the X-ray are performed in a weight-bearing position to demonstrate joint space narrowing [16,17]. Other X-ray findings include osteophyte formation [16,17], subchondral bone sclerosis, as well as erosions and misalignment [16]. Similarly, a CT scan can be used to assess cartilage loss as well [16].

MRI scans, particularly T2-weighted or proton-density weighted sequences with fat suppression, enable the best characterisation of osteoarthritis. The presence of fluid and oedema surrounding the affected joint, along with a history of pain within the same joint, strongly support osteoarthritis as the diagnosis [16]. Routine use of imaging, particularly MRI, to make a diagnosis is discouraged as it drives up rates of unnecessary surgery [20].

### 1.3. Nociceptive, Neuropathic, and Nociplastic Pain

Osteoarthritis pain has nociceptive, neuropathic, and nociplastic components [21,22,23] (See Figure 2). Nociceptive pain indicates ongoing joint inflammation and surrounding tissue damage [24], while neuropathic pain indicates a degree of nerve damage [22,24]. Nociceptive pain in osteoarthritis has been attributed to the sensitisation of peripheral nociceptive receptors in the synovium and subchondral bone [25]. Pain associated with osteoarthritis is often characterised by intense, intermittent pain on a background of persistent aching pain [26], sometimes associated with hyperalgesia and resting pain [27]. Pain due to KOA can refer to the upper tibia and around the knee joint itself [28].

Neuropathic pain is often due to a lesion within the somatosensory system. It is characterised as burning or electrical shooting pain, which is often triggered by stimulations that are not considered painful, such as light touch. Neuropathic pain is often debilitating, affecting patients’ quality of life, as symptoms are often chronic and less responsive to common pain medications [30]. Damage to the nerves, which normally supply the subchondral bone, has been implicated in the pathogenesis of neuropathic pain in osteoarthritis [31]. Additionally, abnormal activity of the damaged nerves may contribute to pain [32]. About one-quarter to one-third of patients with KOA have neuropathic symptoms [32,33,34].

Nociplastic pain has been used to describe chronic pain states that have no obvious nociceptive or neuropathic component [35]. Nociplastic pain has been defined as pain due to impaired function of sensory pathways within the peripheral and central nervous system, resulting in increased pain sensitivity [35]. Nociplastic pain is often difficult to diagnose, given the lack of consistent physical findings and pathology results to support the pain complaint [35]. Nociplastic pain is often described as aching or shooting pain, similar to both nociceptive and neuropathic pain [35]. Additionally, nociplastic pain can be associated with hyperalgesia and dysaesthesia [35]. Nociplastic pain is often not relieved by conventional osteoarthritis management, such as the use of non-steroidal anti-inflammatory drugs, corticosteroids, and knee replacements [33]. Concurrent fibromyalgia and pain resulting from a nociplastic pain syndrome may affect the efficacy of knee and hip osteoarthritis management [36].

PainDETECT questionnaires are potentially useful in assessing neuropathic pain and differentiating neuropathic pain from nociceptive pain [24]. Though painDETECT was initially developed to characterise the neuropathic elements of lower back pain [37,38], it has since been validated for the diagnosis of neuropathic pain in chronic pain conditions [32]. PainDETECT predicts the probability of a neuropathic component in a patient’s pain, enabling better characterisation and individualisation of therapy for patients [24]. Higher painDETECT scores indicated greater neuropathic components to their pain [28]. Patients with higher painDETECT scores also had more cerebral activity on functional MRI with punctate stimuli [27,32]. Other measures for neuropathic pain include the self-report Leeds Assessment of Neuropathic Symptoms and Signs scale [31,38], Neuropathic Pain Questionnaire, and Douleur Neuropathique with four questions [31]. The assessment scoring of osteoarthritis-specific pain will be discussed in further detail below.

### 1.4. Pain Sensitisation

Pain sensitisation is due to either increased peripheral nervous system signal transmission, resulting in peripheral sensitisation or due to intensification of nervous signals within the central nervous system, resulting in central sensitisation40. This can lead to hyperalgesia, allodynia, and pain beyond the region of pathology [39]. Previtali et al. (2022) [39] found that pain sensitisation in KOA is common, with its prevalence ranging from 10% to 56% depending on the method of assessment, with up to 20% of all patients with KOA in their study being impacted by pain sensitisation [39].

Peripheral sensitisation in KOA occurs when nociceptive receptors of the knee tissue become sensitised during inflammation in osteoarthritis [40], whereas continuous nociceptive signals from damage to the synovium, subchondral bone, and joint capsule drive central sensitisation in KOA [25,32]. KOA leads to increased transmission and receptiveness of nociceptive signals at the dorsal horn, leading to hyperalgesia and central sensitisation [27]. Psychological distress, such as anxiety and depression, has been thought to play a part in central sensitisation and neuropathic pain in KOA [32,34,36]. Peripheral and central sensitisation can be assessed using Quantitative Sensory Testing, as described below.

## 2. Concept of Flares; What Causes Them and How to Assess Them

### 2.1. Pain Flares in KOA

Pain exacerbations, or pain flares, are common in KOA [41]. These pain flares are distressing, happen with unpredictable intensity, duration, and frequency and can cause extreme functional disability in those who experience them [26]. Patients describe pain flares as rapid onset, sharp exacerbations of their baseline level of pain, which negatively impact their life. Additionally, it is postulated that flare-ups have the potential to negatively impact disease progression in KOA [41]. Flares occur in the earlier phases of KOA, but in later phases of the disease, they become longer, more intense, less predictable, and more distressing. This phenomenon is in line with the current conceptualisation of KOA as an acute, chronic disease and with the “model of diminishing reserve” of the joint and its lack of capacity to return to baseline as the disease progresses [42].

Approximately 23–32% of individuals with KOA experience pain flares, with nearly 70% of flares being unpredictable in nature [43,44]. Previous research has demonstrated that individuals with different forms of osteoarthritis experience up to 2.4 flares per year, with flares typically lasting 3–8 days [43].

### 2.2. Susceptibility to Flares

The susceptibility to flares is probably due to distal causes and proximal triggers [45]. The distal causes include obesity, disordered biomechanics, and injury. Proximal triggers include insults that have the potential to cause an increase in joint stress and repetitive microtrauma. These joint-stress-related triggers include an increase in physical activity, recent knee injury, squatting or kneeling, lifting heavy loads, climbing ladders, knee buckling, and footwear use. Other proximal triggers have the potential to affect pain threshold and pain perception. These triggers are poor sleep and psychosocial factors, such as poor mood, negative affect, and passive coping [46,47,48,49]. Despite patient-reported anecdotes of the impact of weather, particularly humidity, on pain flares, the effect of weather on pain flares is contentious [50]. It has also been postulated that vitamin D deficiency may be associated with osteoarthritis pain flares, as well as the progression of KOA, as vitamin D is thought to stimulate proteoglycan synthesis [51]. Despite multiple hypotheses, and extensive research, the exact mechanism by which these risk factors cause flares is not well established.

### 2.3. Assessment of Pain Flares

There have been several attempts to develop a definition and delineate osteoarthritis pain flares for research purposes, but a formal medical definition has not been identified [52,53,54]. The first attempt by Marty et al. (2009) [53] developed a scoring system which assigned a weighted point system to features present in flares, with a score ranging from 0 to 14 and a ROC-derived cut-off of 7. These features incorporated in this score include morning stiffness for greater than 20 min, nocturnal awakening due to pain, knee effusion, joint swelling, limping, and warmth over the affected knee. Hawker et al. (2008) [55] developed a multidimensional Intermittent Constant Osteoarthritis Pain Score (ICOAP), an osteoarthritis-specific measure, which examines the pain experience in both hip osteoarthritis and KOA. This 11-item tool considers pain intensity, frequency, and impact on mood, sleep, and quality of life and gives a total pain score from 0 to 44. It considers the constant and intermittent pain scores, the constant pain subscale from 0 to 20, and the intermittent pain subscale from 0 to 2458.

More recently, the Outcome Measures in Rheumatology (OMERACT) flares in osteoarthritis working group identified pain, stiffness, swelling, psychological aspects (including low mood and irritability), and impact as the five domains which characterise a pain flare in both hip osteoarthritis and KOA [56]. In addition, Traore et al. (2022) [54] from this working group developed FLARE-OA, a new 19-item self-reported questionnaire to assess the occurrence and the severity of flares in both knee and hip osteoarthritis [54]. Although these assessment tools are used in research, they are yet to be established for use in clinical practice.

### 2.4. Preventing or Reducing Flares

Identifying and assessing pain flares is necessary to institute remedial measures. Reducing the impact of pain flares, and anticipating daily life triggers which cause these flares, creates an opportunity for individuals (and their health care providers) to implement personalised pain prevention and management strategies. However, the most recent scoping review has highlighted the lack of robust evidence for the management of pain flares [57]. The significance of studies which advocate retro walking (walking backwards) and modified “rescue” exercises during flares is unclear [58,59]. The overarching principle is to support individuals to self-manage their flares and to assist in minimising the impact of flares using ice, the pacing of activities, pain relief, and walking aids or braces to minimise the joint load. Though exercise has the potential to cause a pain flare, in the long term, exercise-induced flares will most likely reduce with graded exercise.

## 3. Assessment of Pain

The pain intensity of KOA can be determined based on various numerical and visual scales. These self-reported questionnaires may be used to assess patient’s pain and are listed in Table 1 below. An alternative or adjunct to these self-reported questionnaires are measures of location (such as knee pain map) and sensitisation, including functional MRI and Quantitative Sensory Testing (such as pressure pain thresholds and conditioned pain modulation).

### 3.1. Knee Pain Map

A knee pain map is a patient-reported indication of the location of knee pain experienced by the patient [81]. It involves having a pictorial representation of bilateral knees that allows the patient to mark out the location of the pain [81]. The location of the pain can be characterised as localised, regional, or diffuse, depending on the location identified [82]. Elson et al. (2011) [81] found that the knee pain map has fair-to-good test–retest reproducibility with good-to-very-good inter-rater reproducibility and very good intra-rater reproducibility [81]. Thompson et al. (2009) [82] also found that the knee pain map has excellent inter-rater reliability in identifying local and regional pain as well as excellent test–retest reliability [82]. It is easy for patients to understand and complete. However, the knee pain only indicates the location of the knee pain and does not provide any information regarding the frequency and intensity of the pain [81].

### 3.2. Functional MRI

Functional MRI differentiates the areas of the brain that are involved in pain [27]. Pain is associated with activity within the primary and secondary somatosensory cortex as well as the insular and prefrontal cortex [27]. Guo et al. (2021) [83] found that patients with chronic KOA pain had reduced grey matter volume in the bilateral insula and hippocampus, increased functional activity in the left insula and bilateral hippocampus, and reduced functional activity in the left cerebellum, as seen on MRI [83]. The grey matter volume of the left insula and functional activity within the left fusiform gyrus on MRI had a significant association with pain intensity [83]. Pujol et al. (2017) [84] found that in participants with KOA, pressure on the area that is tender showed increased brain activity beyond its cortical representation, involving visual, auditory, and sensorimotor cortex distant to the areas represented by the test area [84]. Pain central sensitisation was also found to be common among individuals experiencing chronic pain from KOA [84,85]. Gwilym et al. (2009) [27] have suggested that brainstem activity is a possible indicator of central sensitisation [27].

### 3.3. Quantitative Sensory Testing (QST)

QST may entail either static (for example, pressure or heat pain thresholds) or dynamic measures (such as conditioned pain modulation) [39]. During exposure to noxious stimuli, participants are asked to indicate when they feel discomfort or pain. Discomfort localised to the affected joint may indicate peripheral sensitisation, whereas discomfort away from the affected joint may indicate central sensitisation [86]. There is a variety of measures that can be used in QST, and we will focus on pressure pain thresholds and conditioned pain modulation as these are the ones most widely used in osteoarthritis.

#### 3.3.1. Pressure Pain Thresholds

The pressure pain threshold is the minimum amount of pressure necessary to elicit pain [87]. Pressure pain thresholds are based on 8 predetermined tender points chosen out of the 18 points included in fibromyalgia [88]. These points include the bilateral trapezius, right second rib, right lateral epicondyle, bilateral knees, and bilateral gluteal [88]. A pressure algometer asserts pressure onto these tender points till the participants indicate the first instance of pain, allowing for the measurement of pain thresholds [88]. Pressure pain thresholds over the joint affected by osteoarthritis reflect hyperalgesia localised within the area of the joint, whereas pressure pain thresholds in regions not affected by osteoarthritis represent widespread hyperalgesia, both of which can be found in patients with osteoarthritis [8]. Jaber et al. (2022) [89] found that pressure pain thresholds were not useful in discriminating the impacts of peripheral from central pain in knee function in participants with KOA undergoing arthroplasty [89]. Participants with KOA had lower pressure pain thresholds, which may possibly indicate that KOA results in generalised mechanical hypersensitivity [89]. Moreton et al. (2015) [38] found that higher painDETECT scores correlated with low-pressure pain thresholds in participants with KOA, indicating that neuropathic pain is possibly associated with central pain sensitisation [38].

#### 3.3.2. Conditioned Pain Modulation

Conditioned pain modulation aims to determine the degree of stimulation the nervous system has in response to noxious stimuli, such as pain. This is achieved by comparing the pain thresholds while at rest (without noxious stimuli) and during noxious stimuli exposure [39]. Conditioned pain modulation is thought to reflect the diffuse noxious inhibitory control mechanism, where a noxious stimulus inhibits the intensity of a different noxious stimulus. Individuals with chronic pain conditions often have less effective conditioned pain modulations [90]. Psychological factors, such as high levels of anxiety and depression, can influence conditioned pain modulation results as well [90,91]. In KOA, conditioned pain modulation responses are often altered [40,92], indicating ongoing central sensitisation of pain.

## 4. Application in Clinical Practice

In clinical practice, a detailed history utilising the biopsychosocial framework mentioned above, including the characteristics of pain, is vital for accurate diagnosis of osteoarthritis, differential diagnosis, and identification of particular characteristics that may help in developing a management plan. This includes questions about pain severity, the presence of neuropathic pain, and pain location to help determine the knee compartment mostly affected by osteoarthritis. Table 2 summarises the minimum information that should be obtained by clinicians related to pain and their relevance.

### 4.1. Role of Pain Assessment Tools

When deciding which pain tool to use in the assessment of patients, it is important to take into account the type of pain patients are experiencing, their cognitive status, as well as the availability of time. When a patient is suspected to be experiencing neuropathic pain, usage of the painDETECT questionnaire is a vital tool to distinguish the type of pain patients experience. It is also easily available online for use. Cognitive status is important to consider as well. Questions within the questionnaires and instructions for patients need to be easily understood. Some of the methods mentioned above, such as the McGill Pain Assessment, are time-consuming, making it impractical for daily practice given the limited time clinicians have with patients. Clinicians should opt to use a pain scale that they are most familiar with, and subsequent assessments of the same patient should be conducted with the same pain scale to allow for easy comparison of the scores.

In day-to-day practice, the usage of QST and functional MRI is impractical due to the limited availability and cost of these tools for assessment. Hence, the usage of these assessment tools will likely be limited to research at the present time.

### 4.2. Role of Imaging and Laboratory Tests

The diagnosis of osteoarthritis is clinical, based on the presence of activity or usage-related joint pain and other symptoms and signs in a population of older adults (see Table 3). Imaging and laboratory tests may be requested to rule out other diagnoses (e.g., autoimmune or crystal-related arthritis) or underlying conditions (e.g., haemochromatosis) in atypical presentations where suspicion of other conditions exists.

Large population-based studies have shown that several MRI lesions, such as osteophytes, cartilage, and meniscal and bone marrow lesions are prevalent in the knees of older adults with and without OA, regardless of pain [98]. Excessive use of imaging, such as MRI, may lead to inappropriate diagnosis and unnecessary interventions for the treatment of lesions that are not contributing to symptoms.

### 4.3. Periarticular Causes of Pain

Lesions of the soft tissue around the joint are common (with approximately 20–40% of persons with knee pain) and can occur in knees with and without OA [99,100]. Examples are tendinitis of the quadriceps and patella, patellar and pes anserine bursitis, and iliotibial band syndrome. Identification and appropriate treatment of these lesions often lead to improvements in pain.

### 4.4. Role of Physical Examination

The physical assessment is an extension of the history taken to further characterise the pain and screen for signs of alternative or associated diagnosis. Prominent inflammatory signs, such as the presence of redness, warmth and effusions, should raise suspicion for inflammatory or septic arthritis. The presence of neuropathic features or pain sensitisation can be suggested by the presence of allodynia or hyperalgesia. While allodynia refers to pain due to a stimulus that normally does not cause pain, hyperalgesia refers to the extreme response to a pain stimulus. Physical examination is also helpful in detecting the presence of other joint derangements, such as ligament or meniscal tears and periarticular lesions.

### 4.5. Compartment-Specific Symptoms—Patellofemoral vs. Tibiofemoral Osteoarthritis

KOA can affect one or more of the three knee compartments, including medial, lateral tibiofemoral, and patellofemoral compartments. Pain in patellofemoral osteoarthritis is typically anterior and is exacerbated by climbing stairs or inclines, particularly going down, and activities involving knee bending, such as squatting, prolonged sitting, and standing up from a sitting position. Tibiofemoral osteoarthritis typically causes pain that is more localised in the medial or lateral aspect of the knee.

### 4.6. Tailoring Treatments to Pain Characteristics

Treatment of osteoarthritis pain should be individualised, considering not only the characteristics of pain but also the impact of the osteoarthritis symptoms on the individual and associated conditions, such as sleep disturbances, fatigue, and depression. Non-pharmacological strategies focusing on exercise, weight management, and psychological and biomechanical interventions, when appropriate, are the mainstay of osteoarthritis treatment for all patients. When pharmacological interventions are being considered, differentiating between nociceptive and neuropathic pain can be helpful to guide treatment. While nociceptive pain would be amenable to common analgesics, non-steroidal anti-inflammatory drugs, and local agents, such as intra-articular glucocorticoid injections, neuropathic pain may require the addition of a centrally acting agent, such as antiepileptics or antidepressants. It is of note that only duloxetine, serotonin, and noradrenaline reuptake inhibitor has been recommended by osteoarthritis guidelines [101] and is more widely used for the treatment of osteoarthritis pain. Pregabalin has been shown to reduce central sensitisation, as well as reduce allodynia and hyperalgesia [102], but there is limited evidence to support its use in osteoarthritis. Management of osteoarthritis-associated pain with opioids did not result in superior pain-related function as compared to non-opioid medications [103].

## 5. Application in Trials

### 5.1. Impact of Co-Administration of Analgesics

The use of rescue of concurrent analgesics may influence patient-reported outcomes in clinical trials independent of the effect of the intervention. The use of rescue medications in the placebo and active intervention groups may result in lower levels of reported pain, making it challenging to disentangle the specific treatment effects being tested and dampening the difference in efficacy of interventions [104]. Additionally, the choice to take rescue medications depends on the patient and is not based on a predetermined indication [104]. However, not allowing the use of concurrent or rescue analgesia may adversely impact the conduct of a trial due to difficulties in recruitment, increased dropouts, and reduced compliance to study protocols. Strategies to overcome this issue include recording the use of other analgesics throughout this study and prescribing the same type of concurrent or rescue analgesia to all participants. Participants are commonly instructed to avoid other forms of analgesia during the trial [105].

Additionally, polypharmacy of analgesics can impact the assessment of pain, such as impacting the measurement of conditioned pain modulation [91]. Multiple analgesics, such as ketamine and opioids, have been found to lower the conditioned pain modulation response [106]. Polypharmacy has also been correlated with a poorer self-perceived state of health [107]; this would result in worse initial reports of pain or health status, impacting the baseline level of symptoms.

### 5.2. Regression to the Mean

The regression to the mean phenomenon in clinical trials or longitudinal cohorts refers to the natural variation in pain scores (or any other outcome that naturally fluctuates over time) that may be interpreted as a treatment effect. Regression to the mean occurs when extremely high or low pain scores tend to return to values closer to the mean. Since having pain above a certain threshold is typically one of the inclusion criteria in osteoarthritis trials, trial participants usually have baseline pain scores that are higher than those of the general osteoarthritis population. Regression to the mean may significantly contribute to improvement from baseline in osteoarthritis trials [108].

Despite having clear and objective ways to assess pain, patient data collected may be inaccurate representations of the efficacy of therapies, resulting in ineffective therapies showing benefit [109]. This can impact clinical trials of osteoarthritis as these trials often involve repetitive assessment of participants’ pain levels. As such, with repeated assessments, the level of pain reported may be lower and not truly reflective of the actual improvement in pain levels. Hence, with the development of new treatment modalities for KOA, regression to the mean should always be taken into consideration. However, this can possibly be resolved by testing new therapies with conventional therapies or placebo, as compared to testing new therapies in isolation. This also highlights the importance of more objective measures, such as imaging, as an additional aid in monitoring the progress of structural damage, as this is not affected by regression to the mean.

### 5.3. Contextual Effects

Participant-reported outcomes may also be influenced by a range of contextual factors, including expectations regarding treatment effects, type of intervention, route of administration, and novelty and cost of the intervention, among others. Placebo controls are used to isolate the specific effect of the intervention from contextual effects and non-specific effects on pain, such as natural history and regression to the mean. However, placebo controls may be challenging in trials of complex musculoskeletal interventions, such as those of a combination of different treatments. Pragmatic trials comparing the intervention against usual care or no intervention are commonly used, although this design does not remove contextual effects. Mediation analysis can be used to overcome this issue and isolate specific treatment effects [110].

There are a number of ways to account for the known placebo effects in osteoarthritis trials [111,112]. These include (1) placebo run-ins, (2) removing participants with high pain variance at baseline pretrial, (3) training staff and participants to improve pain reporting accuracy, neutralise expectations, and decrease external cues that may bias participants’ pain ratings [113], and (4) predict and account for individual patients’ placebo responsiveness [114].

### 5.4. Eligibility Considerations

Decisions regarding eligibility criteria may affect the generalisability of the results, recruitment, and ability to detect treatment effects. Stringent eligibility criteria to define a well-defined, homogeneous population in whom intervention may be more likely to be efficacious is typically used in early efficacy studies. Typically, the exclusion criteria include the presence of other conditions that may affect pain assessments, such as fibromyalgia, concomitant inflammatory arthritis, other local sources of pain or referred pain, concomitant hip or back pain (in KOA trials), and psychological conditions that may influence pain status, such as depression, anxiety, and catastrophising. Recruitment for such trials may be challenging, with increased rates of screen failures due to the specific eligibility criteria. Studies with broader eligibility criteria are usually needed to determine the generalisability of the findings [105]. Participants in trials will never be truly homogenous due to the complexity and variability of individuals. Trials which target a specific narrow population will lead to results that are difficult to extrapolate to the wider community, resulting in limited applicability of clinical trials. By the same token, if assessing the efficacy of an intra-articular injection on local pain, some of these other biopsychosocial characteristics may mitigate against the likelihood of finding an effect. A schematic algorithm providing an example for consideration of the effects in the context of a clinical trial is included (See Figure 3).

## 6. Conclusions

When assessing the pain of patients with osteoarthritis, a holistic approach is of utmost importance in ensuring that every aspect of a person’s distress is addressed. Taking into consideration the types of pain patients are experiencing will enable better-targeted management of symptoms. The use of various self-reported questionnaires and Quantitative Sensory Testing enables better insight into the experiences that patients have, allowing for tailored and personalised medicine. However, the use of QST and functional MRI may remain unavailable for use in clinical practice in the near future. Hence, reliance on the use of pain scales, such as the numerical rating scale, ICOAP, or WOMAC, would likely remain the main assessment of pain due to their ease of access and simplicity. The core features of these successful, widely used pain scales are reliability, consistency, and ease of administration.

While it may be challenging to have a unifying pain scale system for research due to the lack of a gold standard for pain assessment in KOA, understanding the implications of co-administration of analgesia and contextual effects on scoring in pain scales will enable better interpretation and understanding of results. The use of multifaceted pain scales, which consider the quality of life, stiffness, and function, proves that patients with osteoarthritis are experiencing distress beyond pain. Consideration of these principles is critical to improving the care of people with osteoarthritis and enhancing success in clinical trials.

## Figures and Tables

**Figure 1 jpm-13-01139-f001:**
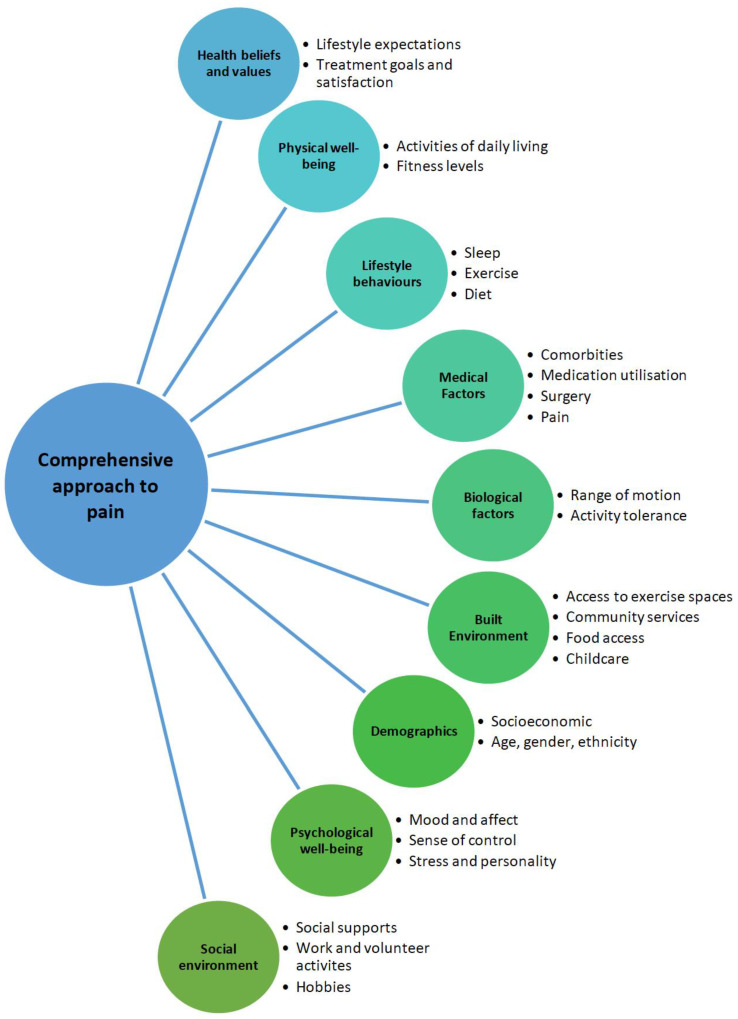
Biopsychosocial framework of assessment adapted from the National Institute for Health and Care Excellence [NICE], 2014 [17].

**Figure 2 jpm-13-01139-f002:**
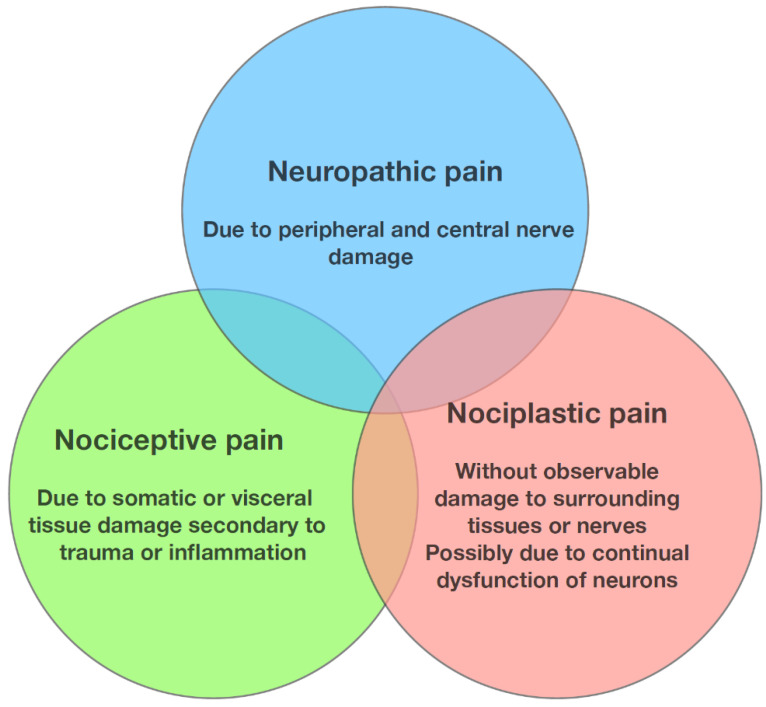
Types of pain in osteoarthritis (adapted from Gebke et al. (2023) [29]).

**Figure 3 jpm-13-01139-f003:**
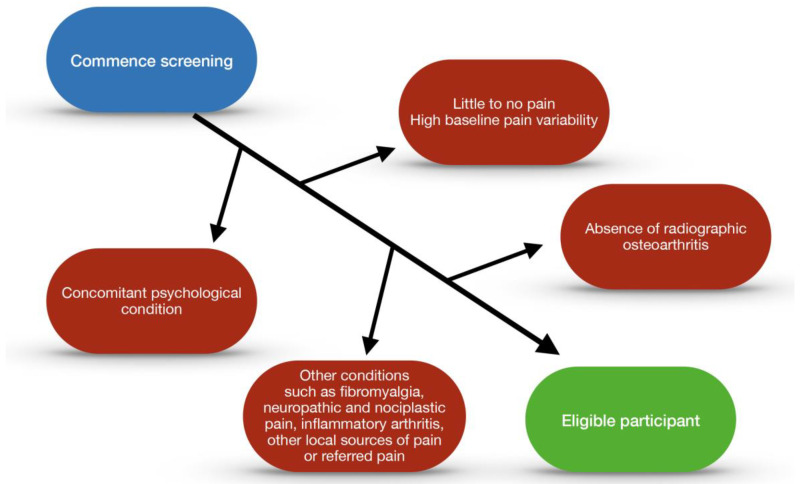
Algorithm for patient exclusion for osteoarthritis trials.

**Table 1 jpm-13-01139-t001:** Self-reporting outcome measures’ advantages and disadvantages.

Self-Reporting Outcome Measures	Method	Advantages	Disadvantages
Numerical rating scale	Pain level is indicated by a number between 2 set numbers, e.g., between 0 and 10 [60]	Feasible [60] Can be administered verbally [60,61] and quickly [61] High test–retest reliability [61] Correlates well to the visual analogue scale in patients with chronic pain conditions [61] A small change in score is considered clinically important [61]	Less distinct discernment of pain intensity [60] Inconsistencies in administration can lead to over and underestimation and inaccuracy [62]
Visual analogue scale	Pain level is marked on a line between two endpoints (no pain and worse ever pain) [60]	Sensitive to treatment effects [60,63] Can be administered quickly [61] Good test–retest reliability [61] Has been shown to be sensitive when detecting changes in chronic inflammatory and degenerative joint pain [61]	Unclear cut-off for clinical significance [60] Time consuming [60] Difficult to understand [60] Cannot be administered verbally [61]
McGill Pain Questionnaire	Consists of a pain rating index where each word has a numerical score, depending on sensory, affective, and evaluative implications of pain, number of words chosen to describe pain and pain intensity (between 1 and 5) [60]	Very extensive [60] Good test–retest reliability within 1 day of testing but poor test–retest reliability within 7 days [61] Able to detect mild pain [61] Differentiates between sensorial and emotional components of pain and has been validated in patients with KOA [6]	Time consuming [61] Difficult to administer and understand [61]
Likert scale	A total of 5 to 7 options between two-end points, e.g., completely better and worse [64]	Practical [64] Easy to understand [64]	Restrictive [64]
Intermittent and constant pain score (ICOAP)	An 11-question questionnaire, each scored between 0 and 4 [65]	Easy to administer [61] Excellent validity for hip and KOA with good test–retest reliability [61] Able to detect changes in osteoarthritis pain intensity in response to pharmacologic therapies [61]	Atukorala et al. (2016) found that ICOAP does not prognosticate pain flares associated with KOA pain flares [65] Does not assess osteoarthritis associated disability [61]
Western Ontario and McMaster Universities (WOMAC) Osteoarthritis index	A 24-question questionnaire about pain, stiffness, and function, scored between 0 and 100, with 100 being the best joint health [66,67]	Good reliability and internal consistency [67,68] Can be self-administered	WOMAC subscale on stiffness is often vague [67]
Knee injury and osteoarthritis outcome score	A 42-item questionnaire about the short and long term impacts of participants’ knee injuries with questions about pain, activities of daily living, function and quality of life, scored between 0 and 100, with 100 being asymptomatic [69]	Self-administered, quick administration, assesses long term outcomes [69] Good test–retest reliability [70,71]	High number of items to be answered, making the assessment time consuming for participants [72]
Lequesne index	An 11-item questionnaire about pain, maximum walking distance, and activities of daily living, with scores ranging from 0 to 8, with a total maximum score of 24, with higher scores indicating worse joint health [68]	Average to good reliability and internal consistency [68]	Needs to be administered by an interviewer [68] Poorer internal consistency and reliability as compared to WOMAC [68]
European health-related quality of life measures (EuroQol)	The 5 questions about pain, mobility, activities of daily living and anxiety [73]	Quick administration and simple to use [73,74] Measures quality of life [73,74] Good reliability and internal consistency [75]	Large ceiling effect [75]
Short Form-36 health survey (SF-36)	A 2-item scale evaluating pain intensity between 6 points ranging from none to very severe [61]	Easy to administer, can be completed quickly with good test–retest reliability [61] Able to detect improvements in pain intensities [61]	Not specific to a disease [61] Hard to distinguish between different intensities of pain, making treatment effect difficult to assess [61]
Neuropathic Pain Scale	A 10-item questionnaire based on 8 qualities of neuropathic pain, such as sharp, hot, cold, sensitive, cold, and itchy [76,77]	Able to differentiate between neuropathic and non-neuropathic pain [76]	Does not include all pain descriptors that patients with neuropathic pain experience [77] Mainly used for monitoring neuropathic pain [78]
Central sensitisation inventory	Questionnaire split into 2 parts. First part involves a 25-item questionnaire with a total score ranging from 0 to 100. Second part involves questions regarding physician diagnosed disorders, such as depression and anxiety [79].	Good test–retest reliability [79], with good sensitivity and specificity [80]	Cut-off scores vary depending on patient sample [80]

**Table 2 jpm-13-01139-t002:** Characteristics of a thorough history of pain in osteoarthritis.

Characteristic	Relevance
Pain location	Local joint pain is typically nociceptive and helps to identify the joint compartment more severely affected (i.e., medial vs. lateral vs. patellofemoral); Pain adjacent to the joint may be related to soft tissue pathology, such as bursitis or tendinopathy of the muscles surrounding the joint; Generalised pain may suggest tri-compartmental joint involvement by osteoarthritis, presence of pain sensitisation, or an alternative diagnosis (e.g., inflammatory arthritis).
Type	Osteoarthritis pain is typically described as stabbing or sharp but can also be described as aching, dull, or throbbing; Tingling, burning, or numbness suggest the presence of neuropathic pain [93].
Aggravating vs. relieving factors	Osteoarthritis pain is typically aggravated by use of the joint and relieved by rest; Pain that worsens at rest should raise suspicion for other conditions, such as inflammatory arthritis and tumours, although the presence of nocturnal pain is common amongst osteoarthritis patients [94] and has been associated with greater osteoarthritis severity [95].
Onset	Osteoarthritis pain is typically gradual; Acute onset of pain should raise suspicion for fractures or other joint derangements, such as meniscal or ligament tears in the appropriate clinical context.
Severity	Pain severity in clinical practice can be assessed by tools, such as the numeric rating scale or self-reported questionnaires, as mentioned in Table 1, and helps in the development of a treatment plan; Severe pain with difficulty of weight bearing should raise suspicion for fracture, tumours, and septic arthritis in the appropriate clinical context.

**Table 3 jpm-13-01139-t003:** Various guidelines for clinical diagnosis of KOA.

Guideline	Criteria for Clinical Diagnosis of KOA
American College of Rheumatology (ACR) [96]	• Presence of knee pain; • With a minimum of 3 of the following: ○ Aged above 50; ○ Stiffness for less than 30 min; ○ Crepitus on active movement; ○ Bony tenderness at joint margins; ○ Bony enlargement; ○ No palpable warmth around synovium.
National Institute for Health and Care Excellence (NICE) [15]	Aged above 45; Joint pain during activities; Morning stiffness not present or lasting less than 30 min.
European Alliance of Associations for Rheumatology (EULAR) [97]	Aged above 40; Presence of knee pain; Momentary morning joint stiffness; Limitations to function; Crepitus; Restricted range of movement; Bony enlargement.

## Data Availability

Data sharing not applicable. No new data were created or analysed in this study. Data sharing is not applicable to this article.

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
