# Peer review of "Assessment of Pain in Osteoarthritis of the Knee"

_jpm, 2023, doi:10.3390/jpm13071139_

Round 1
Reviewer 1 Report
In the manuscript: “Assessment of pain in osteoarthritis of the knee”, the authors discussed about the possible methods of assessing and characterizing pain in patients with KOA using clinical symptoms, pain assessment tools and imaging.
Overall, this manuscript results very interesting, the authors clearly explain the rational of the study and discussed the topic point by point.
However, we would like to invite the authors to clarify some minor points:
1. Please check the check punctuation and spaces;
2. In the first paragraph of introduction the authors said: “Osteoarthritis of the knee (KOA) is a chronic musculoskeletal condition with a multifactorial aetiology. KOA results in chronic knee pain and disability, impacting an individual’s function and quality of life. As pain is a unique, deeply personal and subjective experience, it can be difficult to assess the severity and intensity of the pain that an individual experiences. In this review, we explore the ways to evaluate and quantify pain associated with KOA in a clinically meaningful way in order to better understand the experiences and manage their symptoms more effectively”, please add the relative reference;
3. Among the introduction, the authors should, in general, also briefly introduce the current KOA treatments;
4. Figure 1; the authors should use a bigger font, it is difficult to read in this form;
5. Table 2 and Table 3; please try to re-arrange, they seem a little disordered;
6. Page 13; the authors said: “Use of rescue of concurrent analgesics may influence patient[1]reported outcomes in clinical trials independent of the effect of the intervention”, there is a possible reason? Please add the relative reference;
minor errors of spelling are present
Author Response
Reviewer: 1
Comments to the author:
- Please check the check punctuation and spaces
Response: Thank you for the feedback. We have reviewed the punctuation and spaces in our article and made the necessary changes.
- In the first paragraph of introduction the authors said: “Osteoarthritis of the knee (KOA) is a chronic musculoskeletal condition with a multifactorial aetiology. KOA results in chronic knee pain and disability, impacting an individual’s function and quality of life. As pain is a unique, deeply personal and subjective experience, it can be difficult to assess the severity and intensity of the pain that an individual experiences. In this review, we explore the ways to evaluate and quantify pain associated with KOA in a clinically meaningful way in order to better understand the experiences and manage their symptoms more effectively”, please add the relative reference;
Response: Thank you for your feedback. We’ve included the reference as listed below.
Page 17, Line 616 – 618: Ahluwalia SC, Giannitrapani KF, Dobscha SK, et al. “Sometimes you wonder, is this really true?”: Clinician assessment of patients' subjective experience of pain. Journal of Evaluation in Clinical Practice 2020; 26: 1048-1053. DOI: https://doi.org/10.1111/jep.13298.
- Among the introduction, the authors should, in general, also briefly introduce the current KOA treatments;
Response: Thank you for your feedback. We’ve included the text as written below.
Page 1, Line 34 – 36: Current KOA management comprises of education, weight loss, exercise, analgesia as well as surgical interventions such as knee replacements2.
- Figure 1; the authors should use a bigger font, it is difficult to read in this form;
Response: Thank you for your feedback. We’ve re-designed the figure as included in the edited article.
- Table 2 and Table 3; please try to re-arrange, they seem a little disordered;
Response: Thank you for your feedback. Apologies for the formatting, it was altered during the submission process. We have changed the formatting.
- Page 13; the authors said: “Use of rescue of concurrent analgesics may influence patient[1]reported outcomes in clinical trials independent of the effect of the intervention”, there is a possible reason? Please add the relative reference;
Response: Thank you for your feedback. We have further elaborated on this point as written below.
Page 14, Line 485-490: Use of rescue medications in the placebo and active intervention groups may result in lower levels of reported pain, making it challenging to disentangle the specific treatment effects being tested and dampening the difference in efficacy of interventions102. Additionally, the choice to take rescue medications is dependent on the patient, not based on a predetermined indication102.
Reviewer 2 Report
The authors examined the possible methods of assessing and characterising pain in patients with KOA. Pain assessment was indeed a big challenge in KOA. The logic of the entire manuscript was rigorous, from the epidemiology of OA to the occurrence of pain, assessment, and clinical treatment. Some issues should be addressed.
1. Figures 1, 2, and Supplementary Figure 1 are all from others’ papers, and the author needs to organize and illustrate themselves.
2. The format of the references is quite chaotic, some have doi, some do not, and some have duplicates. The author needs to carefully review and revise.
Author Response
Reviewer: 2
Comments to the author:
- Figures 1, 2, and Supplementary Figure 1 are all from others’ papers, and the author needs to organize and illustrate themselves.
Response: Thank you for your feedback. We have recreated Figure 1. Figure 2 is adapted based on the original article. Supplementary Figure 1 will be removed as it is not a core component of this article.
- The format of the references is quite chaotic, some have doi, some do not, and some have duplicates. The author needs to carefully review and revise.
Response: Thank you for your feedback. All the references have been reviewed. Article DOI have been added where available.
Reviewer 3 Report
The authors explore various methods of assessing and characterizing pain in patients with knee osteoarthritis. And the authors examine current methods of assessing pain in knee osteoarthritis and their application in clinical practice and clinical trials. In this manuscript, the authors also explore the possibility of a personalized pain management program, which has some clinical value. However, I believe that major revisions are needed before they can be considered for publication.
1. The authors do not provide sufficient information about the application of biomaterials in the treatment of osteoarthritis. The authors could refer to the explanations about biomaterials in “Lei Y, Zhang Q, Kuang G, et al. Functional biomaterials for osteoarthritis treatment: From research to application[J]. Smart Medicine, 2022: e20220014” and “Yang L, Sun L, Zhang H, et al. Ice-inspired lubricated drug delivery particles from microfluidic electrospray for osteoarthritis treatment[J]. ACS nano, 2021, 15(12): 20600-20606”.
2. The section on Epidemiology, burden and impact of osteoarthritis deserves a more detailed description of the psychological burden associated with osteoarthritis.
3. I would like the authors to add the characterization experiments of this composite scaffold regarding the degradation properties.
4. I would like the authors to increase the resolution of the images as much as possible, especially for Figure 1 and Figure 2.
5. The authors list a number of methods routinely used in clinical practice to detect and assess osteoarthritis pain. However, I would have preferred that the authors provide more detail on their understanding and opinion of the various methods or suggest an appropriate guideline.
6. The Conclusion section is too brief. It should be a brief summary of the content of the manuscript and should also address the opportunities and challenges that lie ahead. In addition, the author's own views and opinions should be fully presented.
Minor editing of English language required
Author Response
Reviewer: 3
Comments to the author:
- The authors do not provide sufficient information about the application of biomaterials in the treatment of osteoarthritis. The authors could refer to the explanations about biomaterials in “Lei Y, Zhang Q, Kuang G, et al. Functional biomaterials for osteoarthritis treatment: From research to application[J]. Smart Medicine, 2022: e20220014” and “Yang L, Sun L, Zhang H, et al. Ice-inspired lubricated drug delivery particles from microfluidic electrospray for osteoarthritis treatment[J]. ACS nano, 2021, 15(12): 20600-20606”.
Response: Thank you for your feedback. These are very interesting articles which demonstrate some upcoming treatments for osteoarthritis. However, the authors believe that the application of biomaterials in the treatment of osteoarthritis falls outside the scope of this article’s main focus. This article aims at providing a framework for the assessment of pain in osteoarthritis and is germane to all treatments being tested and developed. If we are not understanding the reviewer’s suggestion, please let us know.
- The section on Epidemiology, burden and impact of osteoarthritis deserves a more detailed description of the psychological burden associated with osteoarthritis.
Response: Thank you for your feedback. We have further expanded on the psychological impacts of osteoarthritis.
Page 2, Line 55 - 67: Osteoarthritis is associated with depression, anxiety as well as risk of substance abuse due to the reliance on analgesics for ongoing pain relief10. A retrospective cross-sectional study based on a South Korean national survey involving over 22,000 participants by Lee et al. (2020)10 found a strong association between poor mental health, reduced quality of life and pain from osteoarthritis, with knee osteoarthritis with ongoing knee pain being a significant contributor to depression and stress in females with knee osteoarthritis10. Additionally, Wise et al. (2009)11 found that poor mental health is associated with worse KOA pain, with worsening mental health increasing the risk of pain flares. The authors found that improvements in mental health correlated with improvements in pain11. This further highlights how interlinked mental health and psychological distress are with pain that patients experience.
- I would like the authors to add the characterization experiments of this composite scaffold regarding the degradation properties.
Response: Thank you for your feedback. As mentioned in response number 1 to reviewer 3, the authors have deemed this as beyond the scope of the focus of our manuscript.
- I would like the authors to increase the resolution of the images as much as possible, especially for Figure 1 and Figure 2.
Response: Thank you for the feedback. We have changed Figure 1 and increased the resolution of Figure 2.
- The authors list a number of methods routinely used in clinical practice to detect and assess osteoarthritis pain. However, I would have preferred that the authors provide more detail on their understanding and opinion of the various methods or suggest an appropriate guideline.
Response: Thank you for your feedback. We have included the paragraphs below.
Page 12, Line 396 – 415: Role of pain assessment tools
When deciding which pain tool to use in the assessment of patients, it is important to take into account the type of pain patients are experiencing, cognitive status as well as availability of time. When a patient is suspected to be experiencing neuropathic pain, usage of the painDETECT questionnaire is a vital tool to distinguish the type of pain patients are experiencing. It is also easily available online for use. Cognitive status is important to consider as well. Questions within the questionnaires and instructions for patients need to be easily understood. Some of the methods mentioned above, such as McGill Pain Assessment, are time-consuming, making it impractical for daily practice given the limited time clinicians have with patients. Clinicians should opt to use a pain scale that they are most familiar with and subsequent assessments of the same patient should be done with the same pain scale to allow easy comparison of scores.
In day-to-day practice, the usage of QST and functional MRI is impractical due to the limited availability and cost of these tools for assessment. Hence, the usage of these assessment tools will likely be limited to research at the present time.
- The Conclusion section is too brief. It should be a brief summary of the content of the manuscript and should also address the opportunities and challenges that lie ahead. In addition, the author's own views and opinions should be fully presented.
Response: Thank you for your feedback. We’ve added the following lines to our conclusion.
Page 17, Line 584 – 589: However, the use of QST and functional MRI may remain unavailable for use in clinical practice in the near future. Hence, reliance on the use of pain scales such as the numerical rating scale, ICOAP or WOMAC would likely remain the main assessments of pain due to their ease of access and simplicity. The core features of these successful, widely used pain scales are reliability, consistency and ease of administration.
Page 17, Line 591 – 597: While it may be challenging to have a unifying pain scale system for research due to the lack of a gold standard for pain assessment in KOA, understanding the implications of co-administration of analgesia and contextual effects on scoring in pain scales will enable better interpretation and understanding of results. Use of multifaceted pain scales which consider quality of life, stiffness and function proves that patients with osteoarthritis are experiencing distress beyond pain.
Round 2
Reviewer 3 Report
Accept in present form